# Marine Natural Product Antimycin A Suppresses Wheat Blast Disease Caused by *Magnaporthe oryzae Triticum*

**DOI:** 10.3390/jof8060618

**Published:** 2022-06-09

**Authors:** Sanjoy Kumar Paul, Moutoshi Chakraborty, Mahfuzur Rahman, Dipali Rani Gupta, Nur Uddin Mahmud, Abdullah Al Mahbub Rahat, Aniruddha Sarker, Md. Abdul Hannan, Md. Mahbubur Rahman, Abdul Mannan Akanda, Jalal Uddin Ahmed, Tofazzal Islam

**Affiliations:** 1Institute of Biotechnology and Genetic Engineering (IBGE), Bangabandhu Sheikh Mujibur Rahman Agricultural University, Gazipur 1706, Bangladesh; skpaul_bt@yahoo.com (S.K.P.); moutoshi1313@gmail.com (M.C.); drgupta80@gmail.com (D.R.G.); numahmud_btl@yahoo.com (N.U.M.); rahatsau@gmail.com (A.A.M.R.); mahbub_biotech@yahoo.com (M.M.R.); 2Extension Service, Davis College of Agriculture, West Virginia University, Morgantown, WV 26506, USA; mm.rahman@mail.wvu.edu; 3School of Applied Biosciences, College of Agriculture and Life Sciences, Kyungpook National University, Daegu 41566, Korea; fagunaniruddha@gmail.com; 4Department of Biochemistry and Molecular Biology, Bangladesh Agricultural University, Mymensingh 2202, Bangladesh; hannanbmb@bau.edu.bd; 5Department of Plant Pathology, Bangabandhu Sheikh Mujibur Rahman Agricultural University, Gazipur 1706, Bangladesh; amakanda06@yahoo.com; 6Department of Crop Botany, Bangabandhu Sheikh Mujibur Rahman Agricultural University, Gazipur 1706, Bangladesh; jahmed06@gmail.com

**Keywords:** natural compound, biological control, inhibition, biopesticide

## Abstract

The application of chemical pesticides to protect agricultural crops from pests and diseases is discouraged due to their harmful effects on humans and the environment. Therefore, alternative approaches for crop protection through microbial or microbe-originated pesticides have been gaining momentum. Wheat blast is a destructive fungal disease caused by the *Magnaporthe oryzae Triticum* (MoT) pathotype, which poses a serious threat to global food security. Screening of secondary metabolites against MoT revealed that antimycin A isolated from a marine *Streptomyces* sp. had a significant inhibitory effect on mycelial growth in vitro. This study aimed to investigate the inhibitory effects of antimycin A on some critical life stages of MoT and evaluate the efficacy of wheat blast disease control using this natural product. A bioassay indicated that antimycin A suppressed mycelial growth (62.90%), conidiogenesis (100%), germination of conidia (42%), and the formation of appressoria in the germinated conidia (100%) of MoT at a 10 µg/mL concentration. Antimycin A suppressed MoT in a dose-dependent manner with a minimum inhibitory concentration of 0.005 μg/disk. If germinated, antimycin A induced abnormal germ tubes (4.8%) and suppressed the formation of appressoria. Interestingly, the application of antimycin A significantly suppressed wheat blast disease in both the seedling (100%) and heading stages (76.33%) of wheat at a 10 µg/mL concentration, supporting the results from in vitro study. This is the first report on the inhibition of mycelial growth, conidiogenesis, conidia germination, and detrimental morphological alterations in germinated conidia, and the suppression of wheat blast disease caused by a *Triticum* pathotype of *M. Oryzae* by antimycin A. Further study is required to unravel the precise mode of action of this promising natural compound for considering it as a biopesticide to combat wheat blast.

## 1. Introduction

The wheat blast fungus *Magnaporthe oryzae Triticum* (MoT) pathotype is considered one of the utmost destructive pathogens of the major food crop, wheat [1,2,3,4]. This fungal pathogen has been a threat to three million hectares of wheat crop in some South American countries including Brazil, Argentina, Bolivia, and Paraguay since its first outbreak in Brazil in 1985. Recently, this fearsome wheat killer was introduced into Bangladesh and Zambia, posing a serious threat to global food and nutritional security [3,5,6,7]. The MoT fungus can infect wheat plants at all growth stages, but infection at the heading stage is considered the most destructive [4,8,9]. Infection on the spike at the heading stage blocks the vascular system, resulting in whitehead symptoms with shriveled or no grains in the spike [1,3,4]. Under a favorable environment, yield loss due to wheat blast can be up to 100% [3]. Infected seeds are thought to be the source of inoculum for the long-distance dispersion of this pathogen, while both seeds and airborne asexual spores serve as inoculum for the short-distance dispersion of the pathogen [10,11]. The asexual spore of the fungus is called conidium, a three-celled, hyaline, and pyriform structure, which attaches to the surface of the host by secreting adhesive biochemicals [3,8,12]. After attachment to the host surface, the conidium germinates to form a germ tube. Later on, an appressorium and infection peg is formed to rupture the host epidermis to proceed with the infection process [12,13]. The process of plant tissue invasion is accomplished by the penetrating fungal hypha through the host epidermis and invading the plasma membrane of the host [8,12,13]. Therefore, disruption of any of these asexual life stages eliminates the possibility of pathogenesis by this fungus [14]. 

The use of synthetic fungicides is one of the approaches to control plant disease. Strobilurin (QoI) fungicides, either alone or in combination with other fungicides, have been used to manage wheat blast disease. However, indiscriminate and frequent use of strobilurin (QoI) fungicides has led to the emergence of pathogenic strains resistant to this group of fungicides [15,16,17]. In addition, extensive use of these fungicides can cause serious damage to human health and animals, and can disturb the natural ecosystem [18,19]. Moreover, traditional breeding strategies take a longer time to develop resistant varieties, which often break down under field conditions after a few years due to the development of new pathogen races [15,20]. Due to hexaploidy, the genetic modification of wheat is considered extremely complicated [21]. Therefore, an eco-friendly alternative approach is needed to manage this fearsome disease of wheat.

Biological control may offer a better alternative to the management of plant diseases. Several bacterial genera including *Bacillus*, *Pseudomonas*, and *Streptomyces* are well known for their biocontrol activity [22]. They produce a wide array of antifungal compounds such as lytic enzymes, and antibiotics to suppress growth of pathogens or induce systemic resistance in plants. Among the bacterial genera, the genus *Streptomyces* has received special attention from researchers due to its potential to produce a diverse class of antibiotics [23]. Approximately two-thirds of economically important antibiotics have been isolated from *Streptomyces* spp. that were used for disease management in agriculture [24,25]. The biocontrol activity of these isolated compounds against phytopathogens has been reported by various researchers [25,26,27,28,29,30]. Several studies were conducted using biopesticides as biocontrol agents with specific formulations against plant diseases [31,32]. Moreover, they also produce lytic enzymes such as chitinases and glucanases, which are also used to control phytopathogens [33,34,35]. Either whole cells or metabolites have been used to formulate *Streptomyces*-based fungicides. For example, the mycelia and spores of *Streptomyces* have been used for formulation of Mycostop (containing *S. griseoviridis* K61), Actinovate and Actino-Iron (containing *Streptomyces lydicus* WYEC 108), and RhizovitR (*S. rimosus*) for the control of the foliar and root diseases of various crops [36,37,38,39]. In addition, three secondary metabolites, polyoxin D, streptomycin, and kasugamycin, produced by *Streptomyces* spp. have been marked as foliar fungicides and bactericides [40]. Secondary metabolite-based products provide advantages over live organism-based products due to their higher shelf life and being not amenable to compromising the efficacy due to changes in the environmental conditions.

Antimycin A, a member of antimycin antibiotics isolated from marine *Streptomyces* sp., is composed of acyl and alkyl side chains and a nine-member dilactone ring [41]. Antimycins are known as specific inhibitors of the mitochondrial respiratory chain at the level of complex III [42]. Antimycin A has drawn considerable attention due to its toxicity toward both human and phytopathogenic fungi [42,43,44]. They inhibit the growth and development of the fungi *Rhizoctonia solani* and *Magnaporhe grisea* [45,46]. In a screening of secondary metabolites against a newly introduced destructive wheat blast disease caused by MoT, we found that antimycin A isolated from a marine *Streptomyces* sp. significantly inhibited growth of MoT in vitro. This study aimed to investigate, in detail, the effects of antimycin A on asexual development of wheat blast fungus in vitro and evaluate the control of disease in vivo. Therefore, the specific objectives of this study were to (i) investigate the effect of antimycin A on the suppression of the mycelial growth of MoT; (ii) test its effect on conidiogenesis, conidial germination, and subsequent steps of the asexual development of MoT; and (iii) evaluate the suppression of blast disease at the seedling and heading stages of wheat.

## 2. Materials and Methods

### 2.1. Chemicals

Antimycin A (AMA) (Figure 1) is a chemical compound produced by a marine *Streptomyces* sp. [41]. This pure compound was generously provided by Dr. Hartmut Laatsch of Georg-August University Goettingen, Germany. The fungicide Nativo^®^ 75 WG (a combination of 50% tebuconazole and 25% trifloxystrobin) was purchased from Bayer Crop Science Ltd. Dhaka, Bangladesh.

### 2.2. Culture of Wheat Blast Isolate

A fungal isolate BTJP 4–5 was obtained in pure culture from the field-infected wheat blast samples by picking up a single conidium and then preserving it at 4 °C on dry filter paper, following the method described earlier [47]. For this study, this virulent isolate BTJP 4–5 of MoT was retrieved from storage in potato dextrose agar (PDA) medium and incubated at 25 °C for conidia production [47].

### 2.3. Preparation of Chemical Solution and Conidial Suspension

A stock solution of antimycin A was prepared using a small quantity of dimethyl sulfoxide (DMSO). The preparation of 1, 5, and 10 μg/disk concentrations of this compound was then carried out in distilled water, where the final concentration of DMSO never exceeded 1% (*v*/*v*) in the final solution, which is proven not to affect the hyphal growth or sporulation of MoT. The Nativo^®^ 75 WG concentrations of 1, 5, and 10 μg/disk were prepared in distilled water. Sterilized water with 1% DMSO served as a negative control. The conidial suspension was prepared from 10-day-old culture plates and the spore concentration was adjusted to ca. 5 × 10^4^ conidia mL^−1^, as described by [47].

### 2.4. Fungal Growth Inhibition and Morphological Changes of Hyphae

The hyphal growth inhibition of MoT isolate BTJP 4–5 by antimycin A and Nativo^®^ 75 WG was calculated using the modified disk diffusion method [14,48]. A series of concentrations ranging from 0.005 to 2 μg/disk of antimycin A and Nativo^®^ 75 WG were prepared by dissolving the required amounts in DMSO and water. Nine-millimeter diameter filter-paper disks (Sigma-Aldrich Co., St. Louis, MO, USA) were soaked with the test compounds. The treated disks were placed at a 2 cm distance from one side of 9 cm diameter Petri dishes containing 20 mL of fungal growth media. Five-millimeter diameter mycelial blocks were placed on the opposite side of the filter paper disk. Filter paper disks treated with DMSO followed by evaporation at room temperate functioned as a negative control. The Petri dishes were incubated at 25 °C until the fungal colony fully covered the media surface of the control plates and the experiment was repeated five times. The radial growth of the fungal colony was measured in centimeters with a ruler along with two perpendicular lines drawn on the lower side of each plate. After 10 days of incubation, the data were recorded by measuring the inhibition zone formed by the test compounds and corresponding mycelial growth. The radial growth inhibition percentage (RGIP) (± standard error) [49] was determined from mean values as: RGIP = (Radial growth in control plate − Radial growth in treated plate) ÷ (Radial growth in control plate) × 100.

A Zeiss Primo Star microscope at 40× and 100× (100× was an oil immersion lens) was used to observe the hyphal morphology at the leading edge of the colonies facing the treated and control disks. A Canon DOS 700D digital camera was used to capture images of the disk diffusion experiment. Photographs of the hyphae were captured with a Zeiss Axiocam ERc 5 s microscope.

### 2.5. Inhibition of Conidiogenesis

To induce conidiogenesis, the mycelia of a 10-day-old MoT culture plate were washed to reduce nutrients [3,14,50]. MoT mycelial agar blocks measuring 10 mm were treated with 50 μL of antimycin A and Nativo^®^ 75 WG at 1, 5, and 10 μg/mL and put into Nunc multi-well plates. The same amount of sterilized water was used on the MoT mycelial block with 1% DMSO serving as a negative control. Then, the treated MoT mycelial agar blocks were incubated at 28 °C with >90% RH. Additionally, light and dark periods were adjusted at 14 h and 10 h, respectively. After 24 h, conidiogenesis was examined using a Zeiss Primo Star microscope at 40× magnification. All of the images were captured with a Zeiss Axiocam ERc 5 s microscope and the experiment was repeated five times.

### 2.6. Germination Inhibition and Morphological Modifications of Germinated Conidia

The conidial germination assay was conducted according to the previously described protocol [14,51]. For each treatment, a 100 μL solution of respective concentration was added directly to 100 μL of 5 × 10^4^ conidia mL^−1^ of MoT to make a final volume of 200 μL into a well of a 96-well plate. A glass rod was used to mix the solution immediately, and the solution was incubated at 25 °C. In this experiment, 1% DMSO with sterile water served as a control. A moisture chamber was used to incubate the multi-well plate at 25 °C for 6 h, 12 h, and 24 h in the dark. For each replication, a total of 100 conidia were observed under a Zeiss Primo Star at 100× magnification. Calculations were made of the of germination percentage of the conidia and its developmental process was examined, and photographs were captured with a Zeiss Axiocam ERc 5 s. The time course was repeated five times. The conidial germination percentage (±standard error) was calculated from mean values as: CG% = (C − T)/C × 100, where, CG = conidial germination, C = percentage of germinated conidia in the control samples, and T = percentage of germinated conidia in the treated samples.

### 2.7. Growing of Seedlings

Wheat seeds of Bangladeshi cultivar BARI Gom-26 were surface-sterilized with 70% ethanol for 10 min, soaked in 1.5% active chlorine for 1 hour, and rinsed five times in sterile distilled water (SDW) [52]. Twenty-five seeds were planted in each of the 20 cm diameter plastic pots filled with NPK fertilizer-amended soil. Finally, 20 healthy seedlings per pot were allowed to grow under natural conditions until the seedling bioassay, following the previously described protocol by Gupta et al. [47]. Watering was done as a regular management practice.

### 2.8. Field Evaluation of Antimycin A against Wheat Blast

#### 2.8.1. Preparation of Land and Fertilization

The experiment was set at confined land in the research field of Bangabandhu Sheikh Mujibur Rahman Agricultural University (BSMRAU), Gazipur, Bangladesh. The experimental site was located at 24.09° north latitude and 90.26° east longitude with an elevation of 8.4 m from the mean sea level. The land was well ploughed and cleaned properly by uprooting weeds and stubbles. Well-decomposed cow dung was applied in adequate amounts during land preparation. Chemical fertilizers like nitrogen, triple super phosphate, muriate of potash, and gypsum were applied at the rate of 70-28-50-11 kg ha^−1^ N-P-K-S, respectively [53]. Two-thirds of urea and all other fertilizers were applied at the final land preparation as a basal dose 3–4 days before seed sowing. Immediately after sowing, the plots were lightly irrigated to ensure uniform germination. Irrigation and other intercultural operations were done whenever necessary. The rest one-third of urea was top dressed at first irrigation at 20 days after sowing (DAS). A randomized complete block design (RCBD) was followed for conducting the experiment. 

#### 2.8.2. Seed Sowing and Management of the Plot

The seeds of wheat variety BARI Gom-26 were sown in the first week of December. Seed treatment was carried out with fungicide Vitavex 200 (3 g/kg seed) before sowing. All plots were properly labeled. Irrigation and other intercultural operations were conducted as necessary.

### 2.9. Plant Infection Assay at Seedling Stage

After 14 days of emergence, the pots were covered with sterilized transparent polyethylene bags to maintain humidity and laid in a completely randomized design. Two sets of experiments were conducted for seedling assay. One was preventive and another was curative. In the case of the preventive assay, the seedlings were sprayed with freshly prepared test compounds at the respective concentrations mentioned above and left overnight to dry. The pots were then inoculated by spraying a conidial suspension containing 5 × 10^4^ MoT conidia mL^−1^. Inoculated seedlings were incubated inside sterilized transparent polyethylene bags (>95% relative humidity) at 25 °C and kept in the dark for 24 h after inoculation. The other seedlings were sprayed with a conidial suspension containing 5 × 10^4^ MoT conidia mL^−1^ in the case of the curative assay. The pots were then incubated overnight inside sterilized transparent polyethylene bags (>95% relative humidity) at 25 °C and kept in the dark for 24 h after inoculation for disease development. Then, the seedlings were sprayed with freshly prepared test compounds at the respective concentrations mentioned above. For both preventive and curative assays, the seedlings were then transferred into a growth room operating at 28 ± 1 °C and a minimum of 90% relative humidity with 12 h light per day [54]. In addition, sterilized water was sprayed on the seedlings five to seven times a day to provide a conducive environment for disease development in the growth room conditions. The disease development data were recorded after five days of inoculation. Each treatment was replicated five times.

### 2.10. Infection Assay in Wheat Field at Reproductive Phase

Freshly prepared 1, 5, and 10 μg/disk concentrations of the test compound were sprayed in the respective plots and left overnight to dry; sterilized water with 1% DMSO served as a negative control. Spore suspension was applied in wheat fields just after the flowering stage of the wheat plant. The fungicide Nativo^®^ 75 WG was applied as a positive control and deionized distilled water was applied as a negative control. Before inoculation, the plots were covered with transparent polyethylene sheets to ensure humid condition congenial for spore germination.

### 2.11. Recording of Data, Measurement of Disease Intensity and Severity

At the reproductive phase, the data were collected on total tiller, effective tiller, and infected tiller hill^−1^, full length and infected part of spike, seeds spike^−1^, 1000-grain weight, and grain yield hill^−1^. At the vegetative phase, the data were collected on total seedlings, infected seedlings pot^−1^, full length, and infected part of the leaves. The disease intensity (DI) was calculated using the formula: DI = (Total number of infected plants) ÷ (Total number of plant observed) × 100

Likewise, the blast disease severity assessment was done using a five-scale basis, where % infection means the length of the spike infected by blast. The scales were 0 = no lesions; 1 = 1–25% infection; 2 = 26–50% infection; 3 = 51–75% infection, and 4 = 76–100% length of the spikes infected by blast. The severity of blast was calculated using the formula:$$\mathrm{DS}=\frac{n\times v}{N\times V}\times 100\%$$

DS = disease severity,

n = number of spikes infected by blast,

v = value score of each category attack,

N = number of spikes observed,

V = value of highest score.

### 2.12. Design of Experiment and Statistical Analysis

The experiments in the laboratory and field conditions were performed using a completely randomized design (CRD) and a randomized complete block design (RCBD), respectively, to determine the fungicidal activities of the pure antimycin A compound compared to a standard fungicide. All statistical analyses were conducted using the statistical software package IBM SPSS Statistics 25, and the Microsoft Office Excel 2015 program package. The analysis of means comparison of the treatments was accomplished by Tukey’s honest significance difference (HSD) test (*p* ≤ 0.05). Each treatment was replicated five times and the mean value ± standard error was used in the tables and figures.

## 3. Results

### 3.1. In Vitro Assays of Fungal Growth Inhibition 

The antifungal activity was tested by measuring the inhibition of fungal growth by antimycin A using a plate assay against wheat blast fungus MoT (Figure 2). The compound antimycin A showed a strong inhibition of hyphal growth of the fungus MoT. The mycelial growth inhibition by antimycin A was 62.9 ± 0.42% at 2 µg/disk (Figure 3). Comparative pictures of the suppression of fungal growth by test compounds are shown in Figure 2. 

The bioassay revealed that antimycin A inhibited the growth of mycelia in a concentration-dependent manner. The inhibitory effects of antimycin A increased with the increasing doses, ranging from 0.005 to 2 µg/disk. Based on the lower and higher concentrations, the inhibition percentages of mycelial growth by antimycin A were 9.6 ± 0.38% and 62.9 ± 0.42%, respectively (Figure 3). Antimycin A showed a slightly lower inhibition rate of fungal mycelia than Nativo^®^ 75 WG (Figure 3).

Antimycin A showed extensive inhibition of hyphal growth at 2 μg/disk (62.9 ± 0.42%), followed by 1.5 μg/disk (52.3 ± 1.29%) and 1 μg/disk (50.6 ± 1.19%), which was indicative of a positive correlation of suppression with an increase in concentration. However, this natural product did not show any activity against MoT lower than the dose of 0.005 µg/disk.

The minimum inhibitory concentration (MIC) required for growth inhibition for each inhibitor were obtained. The MIC of Nativo^®^ 75 WG was 0.05 μg/disk, which was 10 times higher than the MIC of antimycin A (0.005 μg/disk). The percentages of fungal growth inhibition at MICs of antimycin A and Nativo^®^ 75 WG were 9.6 ± 0.38 and 22.9 ± 0.64, respectively.

The observation of hyphal growth under a microscope revealed that the hyphae of untreated MoT had polar and tubular growth with smooth, hyaline, regularly branched, septate, plump, and intact structures [Figure 2a(i)]. The MoT-containing Petri dish treated with antimycin A showed irregular growth and frequently increased branch-per-unit length of fungal hyphae. The antimycin A-treated Petri dish also showed rough hyphal cell walls, but displayed ridges with a corrugated existence and irregular swelling of cells [Figure 2b(i)]. Likewise, morphological abnormality also occurred in the case of MoT where the hyphae were close to the disk containing the fungicide Nativo^®^ 75 WG [Figure 2c(i)]. Nevertheless, the altered morphological appearance of MoT by antimycin A was slightly different than those observed with the Nativo^®^75 WG, signifying a possible dissimilar mode of action. Overall, antimycin A is a stronger inhibitor than the commercial fungicide, Nativo^®^ 75 WG.

### 3.2. Antimycin A Block Conidiogenesis in MoT

The generation of conidia asexually from conidiophores is critical for infecting wheat plants with MoT. Both antimycin A and the fungicide Nativo^®^ 75 WG significantly decreased the conidia formation in MoT at 5 and 10 μg/mL when compared to the control. Inhibition increased with an increase in concentration from 1, 5, and 10 μg/mL (Figure 4). MoT failed to develop any conidia at the concentration of 10 μg/mL of both natural antimycin A and the synthetic fungicide Nativo^®^ 75 WG. Microscopic investigation showed broken mycelial tips and a complete lack of conidiophores when the mycelia were treated with 10 μg/mL of both tested compounds. On the other hand, the control dish treated with sterilized water containing 1% DMSO produced 5–6 × 10^5^ conidia/mL.

### 3.3. Antimycin A Alters Conidia Germination and Developmental Transitions of Germinated Conidia 

To determine the germination of conidia and the formation of appressoria of MoT, we used antimycin A and Nativo^®^ 75 WG at 10 μg/mL in multi-well plates. The percentages of germinated conidia and their altered morphology were recorded after 6, 12, and 24 h of incubation (Table 1). Both treatments remarkably reduced conidial germination after six hours of incubation compared to the control. At the same time, 100% conidial germination was observed in water, whereas 44.3 ± 2.33% was observed in plates treated with Nativo^®^ 75 WG. In the antimycin A solution, fungal spore germination was 42.1 ± 0.35%, whereas the commercial fungicide exhibited 44.3 ± 2.33% after 6 h of incubation. At all incubation times (6 h, 12 h, and 24 h), the control treatment supported 100% germination of conidia, developed normal germ tubes, and showed standard mycelial growth at 25 °C in the dark (Table 1 and Figure 5a). At the 10 μg/mL concentration, antimycin A displayed significant adverse effects on both germinations of conidia and impaired the post-germination developmental process of MoT. Overall, antimycin A showed abnormal transitional advancement from one step to the next during the developmental processes of germinated conidia.

After 6 h, in the presence of antimycin A, 26.7 ± 0.41% of conidia germinated with shorter germ tubes compared to the control and 15.4 ± 0.44% conidia lysed. Similar developmental abnormalities were also observed among the germinated spores after 12 h of incubation, which showed 12.6 ± 0.40% normal and 4.8 ± 0.29% with the formation of abnormally elongated germ tubes, while no additional germination was found after 24 h of incubation (Table 1, Figure 5b).

In the presence of Nativo^®^ 75 WG, the germination of conidia was the same at 6 and 12 h of incubation (44.3 ± 2.33), although after 6 h, the conidia showed shorter germ tubes and after 12 h, all of the conidia exhibited normal germ tubes. However, they did not form any appressoria. Similar to antimycin A, the commercial fungicide Nativo^®^ 75 WG also completely suppressed the germination of spores after 24 h (Table 1, Figure 5c). Interestingly, antimycin A yielded abnormally short and long germ tubes and lysed conidia, while the fungicide did not exhibit such changes. Both the natural compound antimycin A and the synthetic fungicide blocked the formation of appressoria that are essential for pathogenesis, suggesting their potential for the control of wheat blast.

### 3.4. Antimycin A Suppresses Wheat Blast Disease at Seedling Stage

The application of antimycin A significantly inhibited the development of blast symptoms in the leaves of artificially inoculated wheat seedlings. In this study, blast lesions on wheat seedlings were treated with antimycin A and Nativo^®^ 75 WG fungicide and were compared with a water treatment control. In the case of antimycin A, the percentages of disease incidence and severity were 16.33 ± 2.19% and 10.67 ± 2.96%, respectively, at the 1 μg/mL preventive dose, whereas 5 μg/mL produced 6.67 ± 0.88 and 3.33 ± 0.88% disease incidence and severity, respectively (Figure 6A(i) and Table 2). Wheat blast caused the highest level (100 ± 0%) of disease intensity and severity (82 ± 4.73%) in the case of the untreated control, whereas the healthy control did not show any blast symptoms (Figure 6B(i) and Table 2).

Additionally, in both of the curative doses, 1 and 5 μg/mL developed 19 ± 1.15 and 8.33 ± 0.67% of plant infection as well as 12.33 ± 2.40 and 5.33 ± 1.20% of leaf infection, respectively. However, the 10 μg/mL preventive and curative doses of antimycin A did not show any blast symptoms on the wheat seedlings (Table 2). However, in both the preventive and curative control measures, 100% suppression of wheat blast was achieved at a 10 μg/mL concentration of antimycin A as well as the commercial fungicide (Figure 6A(d,e) and Table 2).

### 3.5. Suppression of Wheat Blast Disease by Antimycin A at Heading Stage of Wheat under Field Conditions

Wheat blast is predominantly a head disease. To assess the efficacy of the natural product antimycin A in suppressing wheat blast disease in artificially inoculated wheat spikes, we conducted an experiment under field conditions together with a commercial fungicide, Nativo^®^75 WG, at 10 μg/mL. In the field conditions, the application of antimycin A remarkably reduced the incidence of wheat blast (33%) (Figure 7c, Table 3), whereas the disease incidence in the untreated control plot was 87% (Figure 7b, Table 3).

Antimycin A-treated wheat plants had a blast severity of 23.67% compared to 73.67% in the untreated control. The application of both antimycin A (1.95 ± 0.06 gm) and Nativo^®^ 75 WG (2.05 ± 0.13 gm) had a statistically similar, but significantly increased grain yield compared to the untreated control (0.86 ± 0.04 gm). The grain yield in both antimycin A and Nativo treatments were comparable to the negative (non-inoculated, non-treated) check (2.05 ± 0.05 gm) (no artificial inoculation) (Table 3). 

We also recorded 1000-grain weight data for the treatments and found 44.94, 42.73, and 53.57 gm weights for the Nativo^®^ 75 WG, antimycin A, and negative control plot, respectively. These grain yields were significantly higher than those of the untreated control plot (34.43 gm) (Table 3).

## 4. Discussion

Microorganisms are a vital source of novel antimicrobials, as they typically yield toxins to combat different microbes. The biological activity of microbe-derived secondary metabolites can successfully prevent the growth and developmental morphological features of wheat blast spores. In the current study, a natural product, antimycin A, isolated from a marine *Stretomyces* sp., significantly inhibited the mycelial growth and pre-infectional development of wheat blast fungus MoT in vitro. Interestingly, the bioactivity of antimycin A was stronger than that of the widely used commercial fungicide Nativo^®^ 75 WG. Furthermore, this natural compound also suppressed wheat blast disease in artificially inoculated wheat seedlings and spikes, which is comparable to the commercial fungicide Nativo^®^ 75 WG. The bioassay revealed that the inhibition of conidial germination, suppression of appressoria formation, and induction of abnormal mycelial growth by antimycin A are likely to be correlated with blast disease suppression in wheat. To the best of our knowledge, this is the first report of the suppression of the devastating wheat blast fungus by antimycin A extracted from the *Streptomyces* sp., which has the potential to become a fungicidal product or used as a lead compound for controlling the MoT, a killer of wheat.

The biological activities of antimycins are well documented [55,56,57,58,59,60,61]. In an earlier study, a crystalline antibiotic isolated from *Streptomyces kitazawaensis* nov. sp. significantly inhibited the growth of rice blast fungus, the *Pyricularia oryzae Oryzae* pathotype [62]. The antibiotic was identified as antimycin A based on its physical, chemical, and biological properties. Due to its high cost, antimycin A was initially viewed as a promising agent to control rice blast in the greenhouse rather than in the field [62]. Later, a series of experiments in mammalian cells revealed the mode of action of this bioactive secondary metabolite. Antimycin A inhibits mitochondrial electron transport, which disrupts the mitochondrial membrane potentials of mitochondria through the proton gradient across the mitochondrial inner membrane [63,64]. Additionally, antimycin A significantly increases the production of reactive oxygen species (ROS) and causes ATP inhibition as well as glutathione depletion [55,56,57,58,59,60,61]. ROS production and membrane depolarization by antimycin A lead to apoptosis by opening the mitochondrial permeability transition pore. Thus, it releases pro-apoptotic molecules such as cytochrome c into the cytoplasm [64,65,66]. In some instances, antimycin A-induced cell death can be associated with the increased activity of caspase and DNA damage [59,66,67]. Although several researchers reported the activity of antimycin A in mammalian cells, only a few reports are available on the antifungal activity of antimycin A to control plant pathogenic fungi [42,44]. As antimycin A inhibits the electron transport chain in the mitochondria to prevent spore germination, a fungicide can be designed against wheat blast using it as a lead compound.

In this study, we found that hyphal the growth of MoT was inhibited at a lower concentration of antimycin A compared to the commercial fungicide Nativo^®^75 WG. The microscopic observation suggested that antimycin A leads to irregular hyphal growth with frequent branching per unit length of fungal hyphae (Figure 2b(i)). These observations are consistent with a recent observation made by Chakraborty et al. [14], where the authors demonstrated that two *Streptomyces* sp. produced secondary metabolites (oligomycin B and F) and altered the morphological features of MoT hyphae [14]. Some other *Streptomyces*-derived secondary metabolites were also found to cause hyphal growth deformity and inhibition [44]. 

One of the notable findings of this study is the antimycin A-induced swelling of the MoT hyphae [Figure 2b(i),c(i)], which is generally considered as a mode of inhibitory action of a compound against the normal growth and development of pathogenic fungi [68,69]. We examined a series of concentrations of antimycin A ranging from 0.005 to 2 μg/disk. We observed increased hyphal swelling with an increase in concentrations of antimycin A (data not shown). This type of hyphal swelling has been reported in various fungal hyphae by the treatment of polyoxin B [70], fengycin [71], tensin [72], linear lipopeptides, and oligomycins B and F [14,22]. Morphological changes such as profuse branching and swelling of the hyphae of an oomycete pathogen viz. *Aphanomyces cochlioides* by phloroglucinols isolated from *Pseudomonas fluorescence* and xanthobaccin A extracted from *Lysobacter* sp. SB-K88 have been documented [73,74,75,76]. A few studies also evaluated the antifungal properties of antimycin A. Nakayama et al. [62] showed that antimycin A exhibits remarkable inhibitory effects against various fungi, including *Pyricularia oryzae, Alternaria kikuchiana, Gloeosporium laeticolor, Torula utilis, Candida alicans, C. krusei, C. parakrusei*, and *Penicillium chrysogenum* [62]. Rice blast fungus is now considered a *Magnaporthe oryzae* pathotype *Oryzae* (MoO), which is different from the wheat-infecting pathotype MoT. Thus, our work with antimycin A and MoT is the first report of the development of swelling-like structures in the hyphae of this devastating new wheat pathotype.

Most of the pathogenic fungi enter host plants via infecting propagule-like spores or conidia, and the process by which conidia are produced is known as conidiogenesis [77,78]. The suppression of conidiogenesis and germination of conidia reduces the chance of infection by the pathogen. Antimycin A inhibited both conidiogenesis and the conidial germination of MoT in a dose-dependent manner in this study. In addition, other unique antagonistic modes of effect discovered in this study include conidia lysis and abnormally extended hypha-like germ tubes (Figure 5b,c). Consistent with these findings, Chakraborty et al. [14] reported that secondary metabolites from *Streptomyces* sp. suppressed conidiogenesis and the germination of conidia of MoT. Similar work by other investigators revealed that reveromycins A and B from *Streptomyces* sp. inhibited the spore germination of *B. cinerea* and *R. stolonifer* [79]. It has also been reported that during the conidiogenesis and conidial germination process, fungal cells need high energy production in the form of ATP [80]. Antimycin A has been reported to interfere with mitochondrial electron transport by targeting ubiquinol–cytochrome c oxidoreductase, which breaks down membrane potentials and inhibits ATP synthesis. The inhibition of conidiogenesis and conidial germination by antimycin A is likely to be linked with the inhibition of ATP synthesis in MoT cells. A recent experiment with *Fusarium oxysporum* cells provides evidence that an elevated level of ATP is positively associated with conidial germination. Higher ATP production supports the breaking of the dormancy and formation of a germ tube [80]. Therefore, a reasonable justification for the inhibition of the spore germination and hyphal growth of MoT described in this study might be associated with the inhibition of ATP synthesis in mitochondria by antimycin A, indicating that the mode of action of antimycin A for MoT suppression may parallel strobilurin fungicides. More research is needed to unravel the mechanisms involving the interactions of the microbe-derived antibiotic compounds that suppress MoT growth and development. It is also possible that, apart from ubiquinol–cytochrome c oxidoreductase, antimycin A targets an additional protein, Bcl-2 [81]. The Bcl-2 protein is found in the mitochondria of *Colletotricum gloeosporioides*, as well as in other cellular compartments including the ER, and is involved in many stages of the fungal life cycle, including growth, morphogenesis, morpho-pathogenesis, and reproduction. In comparison to the wildtype, Bcl-2 isolates produced 8–10 times more conidia. Furthermore, Bcl-2 isolates are more virulent than wildtype and cause infection faster [82]. The Bcl-2 family has been reported to regulate apoptosis in mammals by controlling the mitochondria efflux of cytochrome c and other apoptosis-related proteins [82]. Whether antimycin A inhibits conidiogenesis in MoT fungus by interfering with the function of Bcl-2 needs to be confirmed by further investigations. 

The hallmark of the findings of this study is that antimycin A remarkably suppressed wheat blast disease in both greenhouse and field conditions, which is comparable to the commercial fungicide Nativo^®^ 75 WG (Figure 6). Herein, the wheat seedlings treated with antimycin A had a lower infection rate than the untreated control (Figure 6), supporting the in vitro results. In contrast, the water-treated healthy control seedlings had a very high percentage of infection, indicating that the efficacy trial was conducted under a conducive environment for higher infection and disease. Products or active compounds that are capable of reducing plant disease under high disease pressure in an experiment should be considered as viable options for future commercial use. In our study, blast lesions did not form on the seedlings treated with antimycin A and Nativo^®^ 75 WG at the highest concentration (Figure 6), indicating the high potential of this antibiotic compound for commercialization in the future to control the wheat blast pathogen MoT. At the heading stage of wheat, we found similar results, where antimycin A significantly inhibited the blast disease development on the artificially inoculated wheat spikes (Figure 7). Excitingly, most of the antifungal effect of antimycin A on the inhibition of wheat blast fungus in vitro was found to be equivalent to or stronger than that of the commercial fungicide at the same concentration. The commercial fungicide Nativo^®^ 75 WG has two main active ingredients, namely, tebuconazole and trifloxystrobin. Tebuconazole acts as a demethylase inhibitor (DMI), which is known as a systemic triazole fungicide. Demethylase inhibitors suppress the biosynthesis of ergosterol, which is a key component of the plasma membrane of certain fungi crucial for fungal growth and development [83]. On the other hand, trifloxystrobin is a strobilurin fungicide that interferes with the respiration of plant pathogenic fungi by inhibiting energy production in the mitochondria, thereby halting the germination of fungal conidia [84]. The adverse effects on conidial germination caused by antimycin A (10 μg/mL) reduced the infection of wheat seedlings by MoT conidia, suggesting a potential role of antimycin A in blast disease suppression. Taken together, both the in vitro and in vivo field tests revealed that antimycin A inhibited mycelial growth, conidiogenesis, and conidial germination, thereby reducing the disease incidence in wheat plants. Our results suggest that antimycin A has a high potential for formulating an effective biopesticide against wheat blast, either using it directly or as a lead compound. Further study is required to elucidate the underlying mechanism of wheat blast disease control by the marine natural product, antimycin A.

## 5. Conclusions

Our experimental findings revealed that a marine natural product, antimycin A, significantly inhibited mycelial growth, asexual sporulation, and the developmental transitions of the conidia of wheat blast fungus MoT in vitro. In vivo seedling assays and field evaluation confirmed that antimycin A was effective in suppressing wheat blast disease in artificially inoculated wheat at both the seedling and heading stages. These assessments proved that this natural compound could be considered as a biofungicide or a lead compound to design a new fungicide to control the destructive wheat blast disease. Further extensive research is also needed to understand the precise mode of action of antimycin A against the devastating fungus, *M. oryzae Triticum*.

## Figures and Tables

**Figure 1 jof-08-00618-f001:**
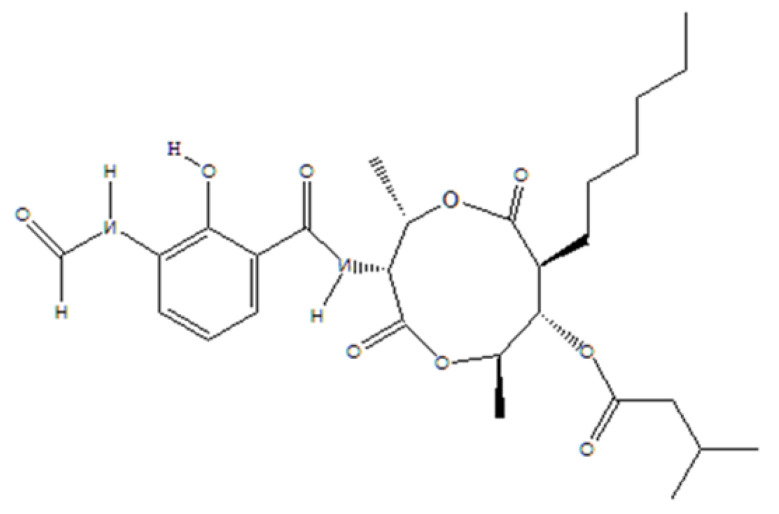
Structure of antimycin A.

**Figure 2 jof-08-00618-f002:**
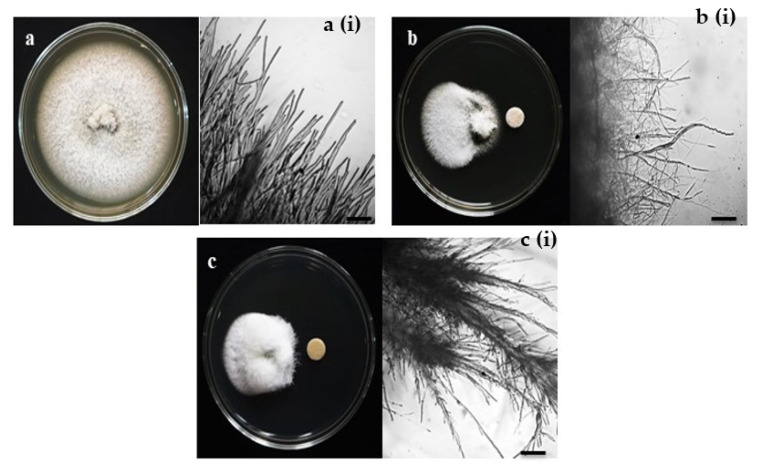
Macroscopic and microscopic view of in vitro antifungal activity of antimycin A and the commercial fungicide Nativo^®^ 75 WG against *M. oryzae Triticum* (MoT) at 20 µg/disk. The macroscopic images are (**a**) control, (**b**) antimycin A, and (**c**) Nativo^®^ 75 WG, whereas, (**a(i)**), (**b(i)**), and (**c(i)**) are microscope images of control, antimycin A, and Nativo^®^ 75 WG, respectively. Bar = 50 μm.

**Figure 3 jof-08-00618-f003:**
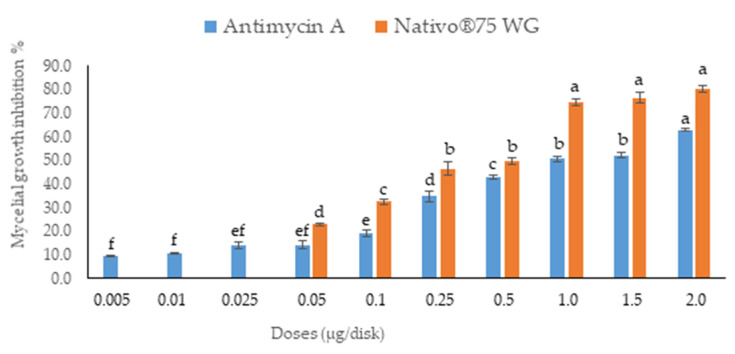
Inhibitory effects of antimycin A and the commercial fungicide Nativo^®^ 75 WG on mycelial growth of MoT in PDA media. The data are the mean ± standard errors of five replicates for each concentration of the compound tested at a 5% level based on the Tukey HSD (Honest Significance Difference) post-hoc statistic. Bars having a common letter are not significantly different at the 5% level of significance.

**Figure 4 jof-08-00618-f004:**
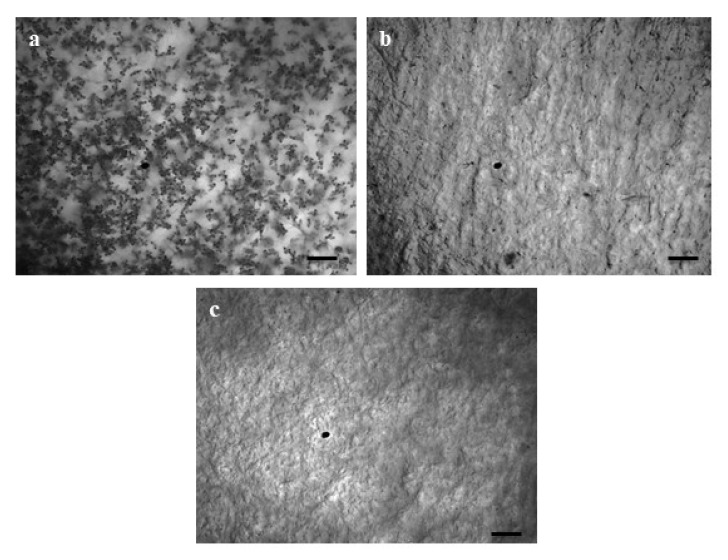
Effects of antimycin A and the commercial fungicide Nativo^®^ 75 WG on the suppression of conidiogenesis of MoT in Nunc multidisc at 10 µg/mL. (**a**) Control, (**b**) antimycin A, (**c**) Nativo^®^ 75 WG. Bar = 50 μm.

**Figure 5 jof-08-00618-f005:**
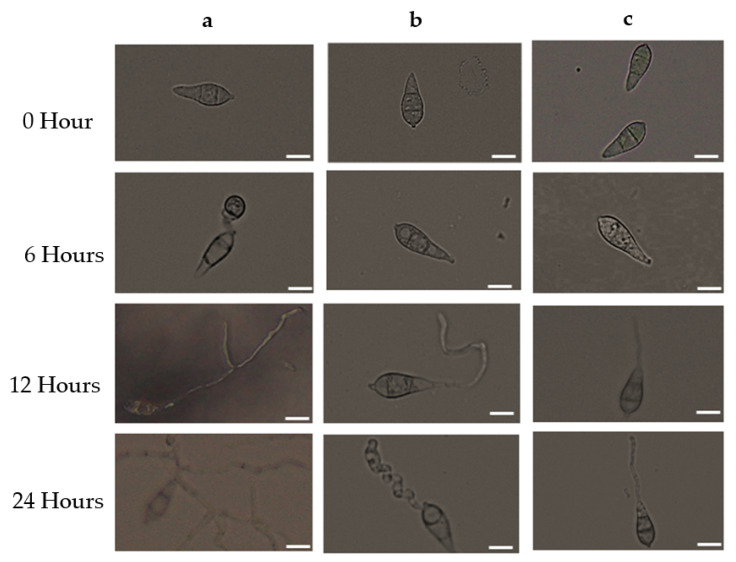
The time-dependent alterations in MoT germination of conidia and subsequent morphological changes in the presence of antimycin A and the commercial fungicide Nativo^®^ 75 WG. Dose of antimycin A was 10 μg/mL. (**a**) Control, (**b**) antimycin A, and (**c**) Nativo^®^ 75 WG. Germinated conidia; short germ tube; elongated germ tube. Bar = 10 μm.

**Figure 6 jof-08-00618-f006:**
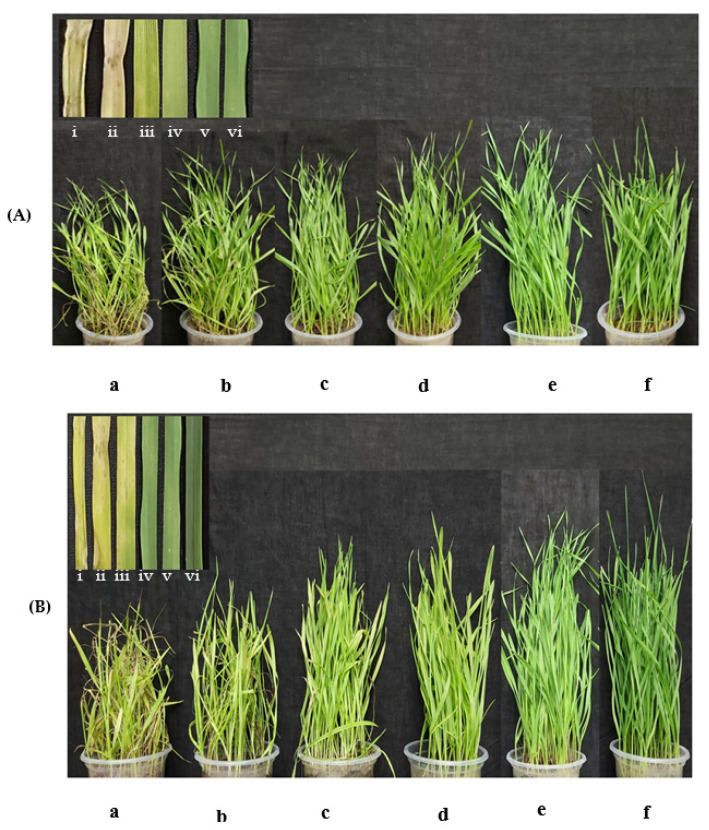
Suppression of wheat blast disease by antimycin A and Nativo^®^ 75 WG. Herein, blast lesions suppressed by different doses of antimycin A, a commercial dose of Nativo^®^ 75 WG, as well as untreated and healthy control wheat seedlings were presented as (**A**) preventive, and (**B**) curative. Both preventive and curative assays included (a) water control + MoT, (b–d) antimycin A + MoT inoculation, (e) commercial dose of Nativo^®^75 WG + MoT inoculation, and (f) non-inoculated, non-treated seedlings. The 1 μg/mL, 5 μg/mL, and 10 μg/mL dose of antimycin A are represented as (b), (c), and (d), respectively. In inset, the representative leaf sample is presented as (i) water control + MoT, (ii–iv) antimycin A at 1 μg/mL, 5 μg/mL, and 10 μg/mL, respectively + MoT inoculation, (v) commercial dose of Nativo^®^ 75 WG + MoT inoculation, and (vi) non-inoculated, non-treated seedlings as a healthy control. Photos were taken 21 days after inoculation.

**Figure 7 jof-08-00618-f007:**
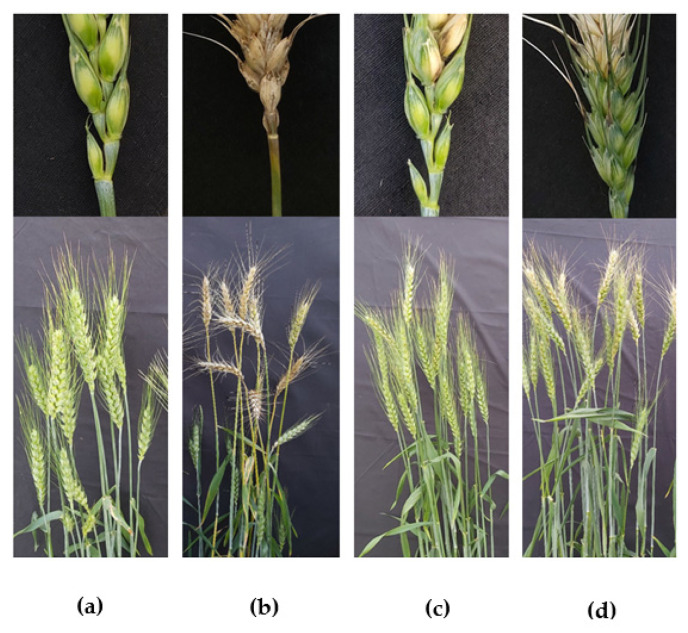
Suppression of wheat blast symptoms with antimycin A and Nativo^®^ 75 WG. Herein, (**a**) non-inoculated, non-treated seedlings as a healthy control, (**b**) water control + MoT, (**c**) antimycin A at 10 μg/mL + MoT inoculation, and (**d**) blast lesions suppressed by commercial dose of Nativo^®^ 75 WG. In inset, clear images of representative spike samples are presented.

**Table 1 jof-08-00618-t001:** Effects of antimycin A and the fungicide Nativo^®^ 75 WG on germination of conidia and morphology of germ tubes and appressoria of *Magnaporthe oryzae* Triticum at 10 μg/mL in vitro.

Treatment	Time (h)	Germination of Conidia, Morphology of Germ Tubes, and Appressorial Formation	
		Germinated Conidia (% ± SE ^a^)	Morphological Change/Developmental Transitions in the Treated Conidia
Water	0	0.0 ± 0.00 ^b^	No germination
	6	100.0 ± 0.00 ^a^	Germination with normal germ tube and normal appressoria
	12	100.0 ± 0.00 ^a^	Normal mycelial growth
	24	100.0 ± 0.00 ^a^	Normal mycelial growth
Antimycin A	0	0.0 ± 0.00 ^c^	No germination
	6	42.1 ± 0.35 ^a^	26.7 ± 0.41% short germ tube and 15.4 ± 0.44% conidia lysed
	12	26.7 ± 0.41 ^b^	12.6 ± 0.40% normal germ tube, 9.3 ± 0.68% short and 4.8 ± 0.29% Abnormally elongated germ tube
	24	0.0 ± 0.00 ^c^	No appressoria, no mycelial growth
Nativo^®^ 75 WG	0	0.0 ± 0.00 ^b^	No germination
	6	44.3 ± 2.33 ^a^	Germinated with a short germ tube
	12	44.3 ± 2.33 ^a^	Normal germ tube
	24	0.0 ± 0.00 ^b^	No appressoria; no mycelial growth

^a^ The data presented here are the mean value ± SE of three replicates in each compound. Means within the column followed by the same letter(s) are not significantly different from those assessed by Tukey’s honest significance difference (HSD) post hoc (*p* ≤ 0.05). Conidia germination percent at different incubation times is not cumulative, but rather at different time intervals in separate experimental units.

**Table 2 jof-08-00618-t002:** Effect of antimycin A in suppression of wheat blast disease development in artificially inoculated wheat seedlings. Means ± standard errors having a common letter are not significantly different at the 5% level of significance.

Parameter	Untreated Control	Healthy Control	Commercial Fungicide Nativo^®^ 75 WG	Preventive (μg/mL)			Curative (μg/mL)		
				1	5	10	1	5	10
% disease incidence	100 ± 0.00 ^a^	0 ± 0.00 ^d^	0 ± 0.00 ^d^	16.33 ± 2.19 ^b^	6.67 ± 0.88 ^c^	0.00 ± 0.00 ^d^	19 ± 1.15 ^b^	8.33 ± 0.67 ^c^	0.00 ± 0.00 ^d^
% disease severity	82 ± 4.73 ^a^	0 ± 0.00 ^c^	0 ± 0.00 ^c^	10.67 ± 2.96 ^b^	3.33 ± 0.88 ^b^	0 ± 0.00 ^c^	12.33 ± 2.40 ^b^	5.33 ± 1.20 ^bc^	0.00 ± 0.00 ^c^

**Table 3 jof-08-00618-t003:** Effect of antimycin A on yield or yield components of the wheat variety BARI Gom-26 under field conditions after artificial inoculation with wheat blast fungus.

Treatment	Grain Yield Per Spike (gm) *	1000-Grain Weight (gm) *	Disease Incidence (%) *	Disease Severity (%) *
Healthy control	2.05 ± 0.05 ^a^	53.57 ± 1.37 ^a^	0.00 ± 0.00 ^c^	0.00 ± 0.00 ^c^
Untreated control	0.86 ± 0.04 ^b^	34.43 ± 0.27 ^c^	87.00 ± 2.91 ^a^	73.67 ± 2.65 ^a^
Antimycin A	1.95 ± 0.06 ^a^	42.73 ± 0.71 ^b^	33.00 ± 1.45 ^b^	23.67 ± 2.31 ^b^
Nativo^®^ 75 WG	2.05 ± 0.13 ^a^	44.94 ± 1.55 ^b^	31.71 ± 2.96 ^b^	23.33 ± 2.95 ^b^

* Any two means ± standard errors with a common letter are not significantly different at the 5% level of significance.

## Data Availability

All data are included in this manuscript as Figures and Tables.

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
