# Peer review of "Marine Natural Product Antimycin A Suppresses Wheat Blast Disease Caused by Magnaporthe oryzae Triticum"

_jof, 2022, doi:10.3390/jof8060618_

Round 1
Reviewer 1 Report
Marine Natural Product Antimycin A Suppresses Wheat Blast
Disease Caused by Magnaporthe oryzae Triticum
The present study is interesting considering that this eco-friendly treatment can protect against Wheat Blast, the study is well described and the discussion and conclusion are aborded in a good manner. I recommend the publication of the present study. Few comments are detailed below.
Keywords: as a suggestion please do not repeat keywords included in the manuscript title
Abstract:
Bioassay indicated that antimycin A suppressed mycelial growth, conidiogenesis, germina- tion of conidia and formation of appressoria in germinated conidia of MoT in a dose-dependent man- ner with minimum inhibitory concentration 0.005 μg/disk.
Comments: in this paragraph, please add the level of inhibition.
Introduction:
Well described the background of the manuscript
Materials and methods
In line 130, adjusted to ca. 5 × 104 conidia mL-1 as described by [45]. ´Please explain what is “ca.”
Results:
Figure 3: please modified the legends for a better understanding, besides add the letters considering the significant effect of treatments
Figure 5. Please modified to plural the time
Table 1. Please change the letters of the significant differences of treatments to superscripts, besides please use the same number of digits for example: 42.1, 0.00 etc
Table 2. same comments as table 1.
Table 3. Please change the letters of the significant differences of treatments to superscripts

Author Response
Reviewer 1 Round 1
Open Review
( ) I would not like to sign my review report
(x) I would like to sign my review report
English language and style
( ) Extensive editing of English language and style required
( ) Moderate English changes required
( ) English language and style are fine/minor spell check required
(x) I don't feel qualified to judge about the English language and style
Yes |
Can be improved |
Must be improved |
Not applicable |
|
Does the introduction provide sufficient background and include all relevant references? |
(x) |
( ) |
( ) |
( ) |
Are all the cited references relevant to the research? |
(x) |
( ) |
( ) |
( ) |
Is the research design appropriate? |
(x) |
( ) |
( ) |
( ) |
Are the methods adequately described? |
(x) |
( ) |
( ) |
( ) |
Are the results clearly presented? |
( ) |
(x) |
( ) |
( ) |
Are the conclusions supported by the results? |
(x) |
( ) |
( ) |
( ) |
Comments and Suggestions for Authors
Marine Natural Product Antimycin A Suppresses Wheat Blast Disease Caused by Magnaporthe oryzae Triticum
The present study is interesting considering that this eco-friendly treatment can protect against Wheat Blast, the study is well described, and the discussion and conclusion are aborded in a good manner. I recommend the publication of the present study. Few comments are detailed below.
Comment
Keywords: as a suggestion please do not repeat keywords included in the manuscript title
Our responses:
Many thanks for the encouraging comments and valuable suggestions for the improvement of the manuscript. We agree with the reviewer’s valuable comment and hence revised the keywords.
Comment
Abstract:
Bioassay indicated that antimycin A suppressed mycelial growth, conidiogenesis, germination of conidia and formation of appressoria in germinated conidia of MoT in a dose-dependent manner with minimum inhibitory concentration 0.005 μg/disk.
-In this paragraph, please add the level of inhibition.
Our response:
Many thanks. According to the reviewer’s valuable comment, we included level of inhibition in the abstract section.
Comment
Introduction:
Well described the background of the manuscript
Our response:
Many thanks for the encouraging comments.
Comment
Materials and methods
In line 130, adjusted to ca. 5 × 104 conidia mL-1 as described by [45]. ´Please explain what is “ca.”
Our response:
Many thanks for your comments. Circa (abbreviated c. or ca.) is a Latin-origin word meaning 'approximately', we removed that from the text to avoid confusion for the readers.
Comments
Results:
Figure 3: please modified the legends for a better understanding, besides add the letters considering the significant effect of treatments.
Our response:
Thank you for your kind suggestion. We modified legends and lettering according to your comments.
Comments
Figure 5. Please modified to plural the time.
Our response:
Thank you for your good comments. We revised our manuscript as you suggested.
Comments
Table 1. Please change the letters of the significant differences of treatments to superscripts, please use the same number of digits for example 42.1, 0.00 etc
Our response:
Thank you for your suggestion. We revised it accordingly.
Comments
Table 2. same comments as table 1.
Our response:
Thank you so much for your valuable suggestion. We revised our manuscript.
Comment
Table 3. Please change the letters of the significant differences in treatments to superscripts
Our response:
We revised our manuscript according to your comments.
Reviewer 2 Report
Reviewers' comments:
The manuscript reports the antifungal activity of antimycin A, a secondary metabolite from a marine Streptomyces sp., and the detailed mechanism of the inhibitory effects of antimycin A against wheat blast pathogen Magnaporthe oryzae Triticum.
Despite the paper shows originality to some extent, it is not able to be published in its present form, and a major revision is suggested,
- As Eilbert et al had reported before (Eilbert, F.; Thines, E.; Anke, H. Effects of Antifungal Compounds on Conidial Germination and on the Induction of Appressorium Formation of Magnaporthe Grisea. Für Naturforschung C 1999, 54 (11), 903–908.), the antifungal activity of antimycin A against Magnaporthe grisea has been evaluated. Therefore, the authors need to emphasize the novelty and originality of their study.
- Why did the authors choose Nativo 75WG as the positive control agent? Is it the prevalent fungicide used for the control of Magnaporthe oryzae Triticum?
- Line 335, the last sentence of Section 3.1, the authors declared that antimycin A is a stronger inhibitor than Nativo 75WG, however, as shown in Fig 3, the inhibition rate of mycelial growth by Nativo 75WG is higher than that of antimycin A in all doses groups.
- Section 3.3, when exploring the antagonistic mechanism of antimycin A on Magnaporthe oryzae Triticum, I recommend that the authors add some activity assays of the enzymes involved in eukaryotic respiration because antimycin A is reported as a respiratory inhibitor.
- Section 3.4, it would be better if the authors illustrate the difference between preventive and curative control measures. It is not explained in the M&M section and is somehow confusing to the readers.
- Did the authors apply the Streptomyces strain directly onto the wheat culture? It will be less expensive and more environmental-friendly than the application of antimycin A itself.
Minor issues,
- Section 2.4, line 139, is it able to evaporate DMSO at room temperature?
- Section 2.11, the second equation, where DI should be DS.
- Figure 2, the caption for the vertical axis is not present correctly.
Author Response
Reviewer 2 Round 1
Open Review
(x) I would not like to sign my review report
( ) I would like to sign my review report
English language and style
( ) Extensive editing of English language and style required
( ) Moderate English changes required
( ) English language and style are fine/minor spell check required
(x) I don't feel qualified to judge about the English language and style
Yes |
Can be improved |
Must be improved |
Not applicable |
|
Does the introduction provide sufficient background and include all relevant references? |
( ) |
( ) |
(x) |
( ) |
Are all the cited references relevant to the research? |
( ) |
(x) |
( ) |
( ) |
Is the research design appropriate? |
( ) |
(x) |
( ) |
( ) |
Are the methods adequately described? |
( ) |
( ) |
(x) |
( ) |
Are the results clearly presented? |
( ) |
( ) |
(x) |
( ) |
Are the conclusions supported by the results? |
( ) |
(x) |
( ) |
( ) |
Comments and Suggestions for Authors
Reviewers' comments:
The manuscript reports the antifungal activity of antimycin A, a secondary metabolite from a marine Streptomyces sp., and the detailed mechanism of the inhibitory effects of antimycin A against wheat blast pathogen Magnaporthe oryzae Triticum.
Despite the paper shows originality to some extent, it is not able to be published in its present form, and a major revision is suggested,
Comment
- As Eilbert et al had reported before (Eilbert, F.; Thines, E.; Anke, H. Effects of Antifungal Compounds on Conidial Germination and on the Induction of Appressorium Formation of Magnaporthe Grisea. Für Naturforschung C 1999, 54 (11), 903–908.), the antifungal activity of antimycin A against Magnaporthe grisea has been evaluated. Therefore, the authors need to emphasize the novelty and originality of their study.
Our responses:
Thank you for your query and suggestion. For the first time, we used Antimycin A in vitro and in vivo to evaluate its performance against wheat blast fungus which is Magnaporthe oryzae Triticum (MoT), whereas Magnaporthe grisea is a rice blast fungus. Rice and wheat blast fungus show host specificity in normal weather conditions and they are pathotype specific. To the best of our knowledge, this is the first report of suppression of the devastating wheat blast fungus by antimycin A extracted from the Streptomyces sp., which has the potential to become a fungicidal product or used as a lead compound to formulate a new product for controlling the MoT, a killer of wheat.
Comment
- Why did the authors choose Nativo 75WG as the positive control agent? Is it the prevalent fungicide used for the control of Magnaporthe oryzae Triticum?
Our responses:
Yes, in Bangladesh Nativo 75WG is recommended for wheat blast control. The chemical can prevent wheat blast in the farmers’ fields. The performance of this fungicide is within the acceptable range from the wheat growers and scientific community in Bangladesh. However, this product is expensive to many resource-poor farmers in the country and creates a threat to human health and the environment. In addition, resistance development in the fungal population against this product warrants alternative active compound-based products.
Comment
- Line 335, the last sentence of Section 3.1, the authors declared that antimycin A is a stronger inhibitor than Nativo 75WG, however, as shown in Fig 3, the inhibition rate of mycelial growth by Nativo 75WG is higher than that of antimycin A in all doses groups.
Our responses:
Thank you for your comment. The reason for this statement is that antimycin A can prevent wheat blast in lower doses than Nativo 75WG. Besides, Nativo 75WG is stronger in higher doses.
Comment
- Section 3.3, when exploring the antagonistic mechanism of antimycin A on Magnaporthe oryzae Triticum, I recommend that the authors add some activity assays of the enzymes involved in eukaryotic respiration because antimycin A is reported as a respiratory inhibitor.
Our response:
Thank you for your valuable comment. Actually, we had no plan for this current project. We will introduce the issue to conduct this enzyme assay in our next project.
Comment
- Section 3.4, it would be better if the authors illustrate the difference between preventive and curative control measures. It is not explained in the M&M section and is somehow confusing to the readers.
Our response:
We revised our materials and methods sections according to your suggestion.
Comment
- Did the authors apply the Streptomyces strain directly onto the wheat culture? It will be less expensive and more environmental-friendly than the application of antimycin A itself.
Our response:
It’s an interesting question. We did not use Streptomyces strain directly onto the wheat culture in this study. However, we have had a good experience on this issue. Our main goal of this project was to evaluate the antifungal performance of antimycin A under in vitro and in vivo conditions which could be used as a lead compound to formulate a new product/fungicide. Microbe-derived secondary metabolites are more desirable than the organism itself by considering shelf-life and other restrictions related to the use of live organisms.
Minor issues,
Comment
- Section 2.4, line 139, is it able to evaporate DMSO at room temperature?
Because of its high boiling point, 189 °C (372 °F), DMSO evaporates slowly at normal atmospheric pressure. Samples dissolved in DMSO cannot be as easily recovered compared to other solvents, as it is very difficult to remove all traces of DMSO by conventional rotary evaporation.
Comment
- Section 2.11, the second equation, where DI should be DS.
Our responses:
Thank you so much for pointing out the editing mistake. We revised it.
Comment
- Figure 2, the caption for the vertical axis is not present correctly.
Our response:
Thank you so much. We corrected our manuscript.
Reviewer 3 Report
Dear Authors
The manuscript with this title"
Marine Natural Product Antimycin A Suppresses Wheat Blast Disease Caused by Magnaporthe oryzae Triticum" is very attractive for readers but before publication I have some suggestions:
- In the introduction, add some biopesticide agents with some formulation.
- please write this line in introduction:
several researches were done about using biopesticides with specific formulations against plant diseases(1,2).
1. F. Fathi, R. Saberi-Riseh, and P. Khodaygan, “Survivability and controlled release of alginate-microencapsulated pseudomonas fluorescens vupf506 and their effects on biocontrol of rhizoctonia solani on potato,” International journal of biological macromolecules, vol. 183, pp. 627–634, 2021.
2. M. Moradi-Pour, Saberi-Riseh. R, Mohammadinejad. R, Hosseini. A, “Investigating the formulation of alginate-gelatin encapsulated pseudomonas fluorescens (VUPF5 and T17-4 strains) for controlling fusarium solani on potato,” International journal of biological macromolecules, vol. 133, pp. 603–613, 2019.
- please write the importance of wheat blast in several regions of world.
- The discussion can be improved
Finally, this manuscript can be published with minor revision.
Author Response
Reviewer 3 Round 1
Open Review
(x) I would not like to sign my review report
( ) I would like to sign my review report
English language and style
( ) Extensive editing of English language and style required
( ) Moderate English changes required
( ) English language and style are fine/minor spell check required
(x) I don't feel qualified to judge about the English language and style
Yes |
Can be improved |
Must be improved |
Not applicable |
|
Does the introduction provide sufficient background and include all relevant references? |
( ) |
(x) |
( ) |
( ) |
Are all the cited references relevant to the research? |
(x) |
( ) |
( ) |
( ) |
Is the research design appropriate? |
(x) |
( ) |
( ) |
( ) |
Are the methods adequately described? |
(x) |
( ) |
( ) |
( ) |
Are the results clearly presented? |
(x) |
( ) |
( ) |
( ) |
Are the conclusions supported by the results? |
( ) |
(x) |
( ) |
( ) |
Comments and Suggestions for Authors
Dear Authors
The manuscript with this title"
Marine Natural Product Antimycin A Suppresses Wheat Blast Disease Caused by Magnaporthe oryzae Triticum" is very attractive for readers but before publication I have some suggestions:
Comment
- In the introduction, add some biopesticide agents with some formulation.
- please write this line in introduction:
several research were done about using biopesticides with specific formulations against plant diseases (1 ,2).
- Fathi, R. Saberi-Riseh, and P. Khodaygan, “Survivability and controlled release of alginate-microencapsulated pseudomonas fluorescens vupf506 and their effects on biocontrol of rhizoctonia solani on potato,” International journal of biological macromolecules, vol. 183, pp. 627–634, 2021.
- Moradi-Pour, Saberi-Riseh. R, Mohammadinejad. R, Hosseini. A, “Investigating the formulation of alginate-gelatin encapsulated pseudomonas fluorescens (VUPF5 and T17-4 strains) for controlling fusarium solani on potato,” International journal of biological macromolecules, vol. 133, pp. 603–613, 2019.
Our responses:
Thank you for your valuable suggestions. We revised our manuscript including the above-mentioned references. Some biopesticide agent and their formulations were included in the manuscript.
Comment
- please write the importance of wheat blast in several regions of world.
Our response:
Thank you. The importance of wheat blast is included in the manuscript.
Comment
- The discussion can be improved
Our response:
Thank you for your valuable comment. We already revised the discussion section of our manuscript.
Comment
Finally, this manuscript can be published with minor revisions.
Our response:
Thank you very much for your positive comments on improving the manuscript and the recommendations for publication.
Round 2
Reviewer 2 Report
All the comments raised by the reviewers were issued properly and thus I suggest this manuscript to be published on Journal of Fungi.